# Cellular Processes in Human Ovarian Follicles Are Regulated by Expression Profile of New Gene Markers—Clinical Approach

**DOI:** 10.3390/jcm11010073

**Published:** 2021-12-24

**Authors:** Błażej Chermuła, Wiesława Kranc, Piotr Celichowski, Bogusława Stelmach, Hanna Piotrowska-Kempisty, Paul Mozdziak, Leszek Pawelczyk, Robert Zygmunt Spaczyński, Bartosz Kempisty

**Affiliations:** 1Department of Gynecology, Division of Infertility and Reproductive Endocrinology, Obstetrics and Gynecological Oncology, Poznan University of Medical Sciences, 33 Polna St., 60-535 Poznan, Poland; blazej.chermula@wp.pl (B.C.); b_stelmach@wp.pl (B.S.); pawelczyk.leszek@ump.edu.pl (L.P.); rspaczynski@yahoo.com (R.Z.S.); 2Department of Anatomy, Poznan University of Medical Sciences, 6 Swiecickiego St., 60-781 Poznan, Poland; wkranc@ump.edu.pl; 3Department of Histology and Embryology, Poznan University of Medical Sciences, 6 Swiecickiego St., 60-781 Poznan, Poland; pcelichowski@ump.edu.pl; 4Department of Toxicology, Poznan University of Medical Sciences, 30 Dojazd St., 60-631 Poznan, Poland; hpiotrow@ump.edu.pl; 5Department of Basic and Preclinical Sciences, Institute of Veterinary Medicine, Nicolaus Copernicus University in Toruń, 7 Gagarina St., 87-100 Torun, Poland; 6Physiology Graduate Program, North Carolina State University, Raleigh, NC 27695, USA; pemozdzi@ncsu.edu; 7Prestage Department of Poultry Science, North Carolina State University, Raleigh, NC 27695, USA; 8Department of Veterinary Surgery, Institute of Veterinary Medicine, Nicolaus Copernicus University in Torun, 1 Lwowska St., 87-100 Torun, Poland

**Keywords:** cumulus cells, granulosa cells, human, gene expression, microarray, epithelialization

## Abstract

In the growing ovarian follicle, the maturing oocyte is accompanied by cumulus (CCs) and granulosa (GCs) cells. Currently, there remain many unanswered questions about the epithelial origin of these cells. Global and targeted gene transcript levels were assessed on 1, 7, 15, 30 days of culture for CCs and GCs. Detailed analysis of the genes belonging to epithelial cell-associated ontological groups allowed us to assess a total of 168 genes expressed in CCs (97 genes) and GCs (71 genes) during long-term in vitro culture. Expression changes of the analyzed genes allowed the identification of the group of genes: TGFBR3, PTGS2, PRKX, AHI1, and IL11, whose expression decreased the most and the group of ANXA3, DKK1, CCND1, STC1, CAV1, and SFRP4 genes, whose expression significantly increased. These genes’ expression indicates CCs and GCs epithelialization processes and their epithelial origin. Expression change analysis of genes involved in epithelization processes in GCs and CCs during their in vitro culture made it possible to describe the most significantly altered of the 11 genes. Detailed analysis of gene expression in these two cell populations at different time intervals confirms their ovarian surface epithelial origin. Furthermore, some gene expression profiles appear to have tumorigenic properties, suggesting that granulosa cells may play a role in cancerogenesis.

## 1. Introduction

The outer part of the ovary is covered by germinal epithelium and somatic granulosa cells, which are modified throughout the ovarian follicular formation process [1,2]. These cells form a direct barrier between the oocyte and the surrounding external environment, creating the proper conditions necessary for oocyte growth and maturation [3]. The process of proper follicular environment formation occurs due to ovarian transformations and endocrine interactions. Paracrine, two-way bidirectional communication between oocyte and granulosa cells, enables antral follicle recruitment and oocyte meiotic resumption [4]. Oocyte growth and maturation, as well as granulosa proliferation and differentiation, are mainly regulated by BMP15 (bone morphogenetic protein 15) and GDF9 (growth differentiation factor 9). Other regulators of these processes are paracrine factors, such as AMH (anti-Mullerian hormone), inhibin, activins, and TGFα (transforming growth factor-alpha), which are produced by the somatic cells surrounding oocytes [5]. An increase of granulosa cell FSH (follicle-stimulating hormone) and IGF1 (insulin-like growth factor 1) surface receptor or oocyte IGF1 receptor activity allows their differentiation into two cell subpopulations: GCs (mural granulosa cells) and CCs (cumulus cells) [6]. GCs are likely to be derived from gonadal ridge coelomic epithelial cells [7]. In the primary follicle, granulosa cells differentiate from a single layer to form a multi-layered COC (cumulus-oocyte- complex) in the mature Graafian follicle, consisting of the corona radiata and cumulus oophorus cells [8].

Granulosa cells are not well described in the current literature. It appears that mesothelial ovarian surface cells that cover the ovary are indicated as pregranulosa precursors [9]. In humans, the morphological evidence indicates that the OSE (ovarian surface epithelium) contributes to the pregranulosa cell population in newly forming follicles. Sawyer, H.R. et al. hypothesize that most (i.e., >95%) of the granulosa cells in newly formed primordial follicles originate from the ovarian surface epithelium [10]. The ovarian surface epithelium maintains the tight GC structure. GCs are characterized by morphological variability, changing their shape from flat, through cuboid, to columnar [11]. Furthermore, it is suggested that fetal OSE can give rise to ovarian granulosa cells [12,13]. Other reports indicate their origin from ovary rete tubules [14] or from centrally located blastema cells [15]. In most cases, studies were performed on animal models; therefore, the analysis of the origin of granulosa cells requires confirmation on human cells. Confirmation of granulosa cells’ origin may help to approximate these cells’ function in the ovarian follicle structure and the entire ovary. These cells are essential for oocyte maturation during folliculogenesis, Graffian follicle ovulation, and corpus luteum formation [16]. The negative aspect of these cells may be the possibility of their transformation into cancer cells [17].

Taking into account the discrepancies in the type of cells from which ovarian tumors arise, the majority of reports indicate their epithelial origin [18]. A thorough analysis of the expression of the CCs and GCs genes may give an answer to how granulosa cells can be involved in ovarian neoplastic processes.

Hence, GCs and CCs etiology still needs to be fully elucidated. Interestingly, based on the activity of specific signaling pathways, OSE stem cells have been characterized to be potentially involved in ovarian carcinogenesis. Wright et al. claim that epithelial carcinogenesis is one of the main causes of ovarian tumors, additionally suggesting that OSE removal significantly reduces the probability of ovarian cancer [19,20].

In recent years, due to the use of expression microarrays, GC and CC transcriptome analysis has become a powerful tool for the improvement of knowledge about the pathways involved in these cells’ development and associated with oocyte growth [21,22]. Considering earlier reports about ovarian epithelial cell differentiation possibilities in GCs or CCs, in the presented research, we have attempted to investigate and explain the genetic basis of these processes.

The objective of the current study was to identify genes whose expression may indicate the OSE origin of GCs and CCs. Additionally, epithelial stem cells exhibit active proliferation of tubal epithelial cells and an increase in the expression of genes involved in tube morphogenesis in vitro [23]. Microarray expression technology was used following previous studies published by Ożegowska et al. [24], which analyze gene expression in porcine oocytes before and after maturation. Involved in epithelial processes, genes expression analysis was carried out on the basis of a 30-day in vitro culture model. Granulosa cells’ long-term in vitro cultivation carried out outside the organism can reveal the primary functions and biological origin of these cells. Additionally, the 30-day in vitro culture of CCs and GCs may reveal the directions in which these cells can differentiate under laboratory conditions [25,26].

The 30-day cell culture protocol allows for an accurate understanding of cell properties in new in vitro conditions. The 0 point (24 h) corresponds approximately to the physiological properties of cells [27], while the following days show the changes that occur in culture. The 7th day defines the short-term culture, the 15th day reflects changes after the first passage, while day 30 is the end of long-term culture [26]. CC and GC long-term in vitro cultures can also show us the direction of their progression in laboratory conditions [16,28]. It also seems interesting to correlate the growth and differentiation of these cells in in vitro conditions with their in vivo behavior.

Long-term in vitro cultures allow the visualization of the primary features of cells [26]. From a clinical point of view, understanding the properties of CCs and GCs at an early stage of their growth may be useful in understanding ovarian primary follicles’ mobilization mechanism during patient stimulation in IVF procedures.

Based on previous studies conducted on animal models [29], an objective was to correlate the growth and differentiation of these cells in in vitro conditions with their behavior and possible influence on ovarian tumor formation. Fallopian tubal epithelial cells may also be mainly responsible for ovarian carcinogenesis [30], and a goal was to elucidate the role of granulosa cells in the process of carcinogenesis.

The main goal of this study was to analyze the expression change of CC and GC genes involved in epithelial processes.

## 2. Materials and Methods

Human ovarian granulosa and cumulus cells long-term in vitro culture and gene expression analysis allowed us to study, on a large scale, a wide group of 168 genes belonging to ontological groups characterizing epithelial cell physiological processes. The main focus was on: “Epithelial cell apoptotic process” and “Epithelial cell migration” ontology groups in GCs and CCs; “Epithelial cell development”, “Epithelial cell differentiation”, “Epithelial cell morphogenesis”, “Epithelial cell proliferation”, “Epithelial to mesenchymal transition”, “Epithelial tube morphogenesis”, “Epithelium development” in CCs; and “Regulation of epithelial cell differentiation” or “Regulation of epithelial cell migration” in GCs. The samples analyzed were isolated on the 1st, 7th, 15th, and 30th day of culture.

### 2.1. Patients Selection

GC and CC cells were obtained from patients undergoing in vitro fertilization (IVF) at the Centre of Diagnosis and Treatment of Infertility at the Division of Infertility and Reproductive Endocrinology, Poznan University of Medical Sciences, Poland. After providing informed consent, the material was obtained from 12 women seeking infertility treatment (mean age 33.67 years ± 1.46 (SEM; standard error of the mean); range 25–40), with normal body mass (mean body mass index (BMI) 21.29 ± 1.46 kg/m^2^), and normal ovarian reserve (mean concentrations in the early follicular phase of the previous cycle: AMH 2.72 ± 1.24 ng/mL and FSH 6.50 ± 1.84 mIU/mL). Infertile women introduced into the study had no previous ovarian surgeries, no other chronic medical conditions, nor endocrinopathies. In addition, patients with PCOS (polycystic ovary syndrome), endometriosis, and POI (premature ovarian insufficiency) were excluded, and only patients with previously identified tubal or male infertility factors were qualified for the investigation. The study also excluded patients with inadequate risk of ovarian stimulation based upon the Bologna criteria for poor ovarian response patients [31].

Ovarian hyperstimulation was induced by the administration of recombinant human follicle-stimulating hormone (rhFSH; Gonal F, Merck sp. z o.o., Poland or Puregon, MSD Polska sp. z o.o., Poland) combined with highly purified human menopausal gonadotropin (hMG; Menopur, Ferring Pharmaceuticals Poland sp. z o.o., Poland) in individually selected doses, following normal standards of care (Cetrotide, cetrorelix 0.25 mg, Merck sp. z o.o, Poland or Orgalutran, ganirelix 0.25 mg, MSD Poland sp. z o.o, Poland). To induce oocyte maturation and ovulation, 9–12 days after the first administration of gonadotropin, patients were injected with chorionic gonadotropin (rhCG; Ovitrelle, 250 ug, Merck sp. z o.o, Poland). Oocytes were collected during transvaginal ultrasound 36 hours after rhCG administration. This research has been approved by Poznan University of Medical Sciences Bioethical Committee with resolutions 1290/18, 558/17, and 196/20.

### 2.2. Patients CC and GC Acquisition

During a standard OPU (oocyte pick up) procedure, cumulus-oocyte complexes were obtained and FF (follicular fluid) with suspended GCs cells was secured. A routine procedure for oocyte preparation for further fertilization by ICSI (Intra Cytoplasmic Sperm Injection) involves its prior denudation. This process involves mechanical and enzymatic (800 IU/mL HYASE-10 X) removal of the surrounding oocyte corona radiata and cumulus oophorus somatic cells forming COC. On average, 10 COCs were obtained from each patient. After complete oocyte denudation, the cumulus cells were collected from individual patients and pooled for further analysis. In the case of follicular fluid previously collected during the OPU procedure, the FF samples were centrifuged for 10 min at 200× *g* to obtain a suspension of GC in the pellet form [25]. Both in the case of CCs and GCs obtained from each patient, separate 30-day in vitro cultures were carried out.

### 2.3. Cell In Vitro Culture

Cells obtained in this way were washed twice in culture medium by centrifugation at 200× *g* for 10 min at RT (room temperature). For each of the 12 patients, a 30-day CCs culture and a 30-day GCs culture were performed. The culture medium consisted of DMEM (Dulbecco’s Modified Eagle’s Medium, Merck KGaA by Sigma-Aldrich, Darmstadt, Germany) supplemented with 2% fetal bovine serum FBS (FBS; Sigma; Merck KGaA), 4 mM l-glutamine (stock 200 mM, Invitrogen by Thermo Fisher Scientific, Inc., Waltham, MA, USA), 10 mg/mL gentamicin (Invitrogen by Thermo Fisher Scientific, Inc.), 10,000 U/mL penicillin, and 10,000 μg/mL streptomycin (Invitrogen; Thermo Fisher Scientific, Inc.). The primary in vitro culture was carried out for 30 days divided into four time intervals. The first is the 0 point (24 h), which reveals the physiological properties of the cells [27] observed in vivo. The following days show the changes in the cultures: the 7th day of in vitro culture determines the short-term culture, while the 15th day shows the effects of the first passage. On day 30, it is possible to note the changes that occurred at the end of the long-term in vitro culture. CCs and GCs were counted using the “Neubauer improved” counting chamber (ISO LAB Laborgerate GmbH, Wertheim, Germany; DIN Certificate EN ISO 9001). The cells were cultured for 30 days at 37 °C and 5% CO_2_. After the cells reached 90% confluence, they were separated from the bottom using 0.05% trypsin-Ethyleno Diamine Tetra Acetic (trypsine—EDTA Invitrogen; Thermo Fisher Scientific, Inc., Waltham, MA, USA) for 1–2 min and counted using an ADAM Cell Counter and Viability Analyzer (Bulldog Bio, Portsmouth, NH, USA). In every three days of culture, the culture medium was changed. Cells were harvested on days 1, 7, 15, and 30 of culture. Before the acquisition of cells, photographic documentation of their shape was made using an inverted microscope (Olympus IX73, Tokyo, Japan). Sample criteria for further analysis were 95% viability; each culture was maintained independently. The viability was assessed by using the ADAM CCVA analyzer. In each of the patients, in each of the four time intervals, two pools of cells (CCs and GCs) were obtained and subjected to the RNA isolation procedure.

### 2.4. RNA Isolation

Total RNA isolation was based on the modified Chomczyński and Sacchi method [32,33]. RNA was isolated from the CCs and GCs after 1, 7, 15, and 30 days of culture. The isolated RNA was resuspended in 100 μL pure water, with the NanoDrop spectrophotometer (Thermo Scientific, Warsaw, Poland) used to measure its purity and concentration. For further analysis, samples with a 260/280 absorbance co-efficiency greater than 1.8 were used.

### 2.5. Microarray Expression Analysis and Statistics

Two total RNA (100 ng) samples isolated and pooled from CC and GC cells were subjected to two rounds of sense cDNA amplification (Ambion^®^ WT Expression Kit, Ambion, Austin, TX, USA). The obtained cDNA was used for biotin labeling and fragmentation using Affymetrix GeneChip^®^ WT (Affymetrix, Santa Clara, CA, USA) Terminal Labeling and Hybridization. Biotin-labeled fragments of cDNA (5.5 μg) were hybridized to Affymetrix^®^ Human Genome U219 Array strips (48 °C/20 h). For both human cumulus and human granulosa cell samples, the same array type was used. Microarrays were washed and stained according to the technical protocol, using the Affymetrix GeneAtlas Fluidics Station. The array strips were scanned employing the Imaging Station of the GeneAtlas System. The preliminary analysis of the scanned chips was performed using the Affymetrix GeneAtlas^TM^ Operating Software. The quality of gene expression data was confirmed according to the quality control criteria provided by the software (http://www.affymetrix.com/support/technical/byproduct.affx?product=geneatlas; accessed on 19 September 2019). The obtained CEL files were imported into the downstream data analysis software (http://www.affymetrix.com; accessed on 19 September 2019) [34,35].

All of the presented analyses and graphs were performed using Bioconductor and R (v3.8.3) programming languages. Each microarray experiment was analyzed separately. The Affy package (3.12) [36] was used to perform the Robust Multiarray Averaging (RMA) algorithm to correct background, normalize, and summarize results. To determine the statistical significance of the analyzed genes, moderated t-statistics from the empirical Bayes method were performed. The obtained *p*-value was corrected for multiple comparisons using Benjamini and Hochberg’s [37] false discovery rate. These calculations were performed by means of the Limma package (3.12) [38]. The comparisons and statistics were performed between the first 24 h of the experiment and the rest of the samples.

Differentially expressed genes were subjected to selection by examination of ontology groups involved in epithelial processes. The differentially expressed gene list was uploaded to the DAVID software (Database for Annotation, Visualization, and Integrated Discovery) [39], where genes belonging to the terms of all three GO (Gene Ontology) domains were extracted. Expression data of these genes were also subjected to a hierarchical clusterization procedure, and their expression values were presented as a heat map.

Genes with a fold change higher than abs (2) and with corrected *p*-value lower than 0.05 were considered as differentially expressed. This set of genes consists of 2278 different transcripts.

Subsequently, the GO BP (Gene Ontology Biological Process) terms from both experiments were used to search for differently expressed genes whose expression changed in both investigated cell cultures.

DAVID (Database for Annotation, Visualization, and Integrated Discovery) software (v.6.8) was used for the extraction of GO BP that contains differentially expressed transcripts. Up and down-regulated gene sets were subjected to DAVID searching separately, and only gene sets where adj. *p*-values were lower than 0.05 were selected.

Subsequently, the relationship between the differentially expressed genes was investigated using the GOplot package, which was also used to calculate the z-score [40]. This z-score does not refer to the standard score from statistics but is an easy-to-calculate value to give you a hint if the biological process (molecular function/cellular components) is more likely to be decreased (negative value) or increased (positive value). It is calculated as follows: the number of up-regulated genes minus the number of down-regulated genes divided by the square root of the count. Whereas up and down are the number of assigned genes up-regulated (logFC > 0) in the data or down-regulated (logFC < 0), respectively. This information allowed estimating the change of course of each gene-ontology term.

Interactions between differentially expressed genes belonging to the gene ontology groups of interest were investigated using STRING10 software (Search Tool for the Retrieval of Interacting Genes) [41]. Genes were used as a query for an interaction prediction. The search criteria were based on text mining, co-expression, and experimentally observed interactions. The analyses gene/protein interaction network reflected the strength of the interaction score.

### 2.6. RT-qPCR Analysis (Reverse Transcription Quantitative PCR)

RT-qPCR analysis was performed to confirm results obtained by microarray analysis. The individual samples have been used in microarray validation. To eliminate technical errors related to the application of reagents to a 96-well plate, each biological repetition was performed in three technical repetitions. One gene with the decreased expression (TGFBR3) change and four genes with the increased expression (STC1, CCND1, DKK1, ANXA3) have been validated in CCs. In addition, GC-specific genes were validated: genes with decreased expression; PTGS2, PRKX, AHI1, IL11, and increased expression; CAV1, SFRP4. Each analysis was performed in triplicate technical repetitions. For reverse transcription, the SABiosciences kit (Frederick, MD, USA; RT2 First Stand kit-330401) and the 96-well Verlerimer thermocycler were used. In each reaction, 1 µg of RNA transcript was used for amplification. The amplification parameters were: preincubation at 37 °C for 30 s; 3-step amplification (95 °C for 15 s, 58 °C for 15 s, 72 °C for 15 s) for 45 cycles; melting (95 °C for 60 s, 40 °C for 60 s, 70 °C for 1 s, 95 °C for 1 s); cooling at 37 °C for 30 s. Gene expression was analyzed using the 2−ΔΔCq method [42]. Real-time PCR was performed using a Light Cycler^®^ 96 (Roche Diagnostic GmbH, Germany), Master Mix RT2 SYBR^®^ Green ROX ™ qPCR (Qiagen Sciences, Gaithersburg, MD, USA), and sequence-specific primers (Table 1). Gene expression was calculated using the RQ (relative quantification) method, using ACTB (β-actin), HPRT1 (hypoxanthine 1 phosphoribosyl transferase), and GAPDH (3-phosphate glyceraldehyde dehydrogenase) genes as references. Relative quantification was used for gene expression analysis. The RT-qPCR primers were designed by using Primer3Plus software (version 0.4.0; Whitehead Institute for Biomedical Research, Massachusetts Institute of Technology, Cambridge, MA, USA). RT-q PCR statistical analysis was performed with the use of the Real Statistics Resource Pack for MS Excel 2016 (Microsoft Corporation, Redmond, WA, USA).

## 3. Results

The dynamic changes in gene expression were investigated by the transcriptomic profile on the 1st, 7th, 15th, and 30th day of human CC and GC cultures. Using the Human Genome, U219 Array transcripts were evaluated to reveal differential expression. The array used in this study enabled the analysis of 49,533 targets on the microarray, which 22,480 were annotated genes, and 2278 had significantly altered expression.

The DAVID software analysis showed that differentially expressed genes belong to 657 Gene ontology terms for human cumulus cells and 582 Gene ontology terms for human granulosa cells. A total of 98 differentially expressed genes from human cumulus cells that identifies “epithelial cell apoptotic process”, “epithelial cell development”, “epithelial cell differentiation”, “epithelial cell migration”, “epithelial cell morphogenesis”, “epithelial cell proliferation”, “epithelial to mesenchymal transition”, “epithelial tube morphogenesis” and “epithelium development” GO BP terms undertook a more detailed examination. Similarly, 72 differentially expressed genes from human granulosa cells that belong to “epithelial cell apoptotic process”, “regulation of epithelial cell differentiation” and “regulation of epithelial cell migration” GO BP terms were evaluated.

Among CCs, TGFBR3 (transforming growth factor-beta receptor 3), HMGB1 (high mobility group box 1), and ARF6 (ADP ribosylation factor 6) genes were identified as genes whose expression decreased significantly. The second CCs group, containing ANXA3 (annexin A3), DKK1 (dickkopf WNT signaling pathway inhibitor 1), CCND1 (cyclin D1), and STC1 (stanniocalcin 1) genes, had the largest increase in expression during culture. Differentially expressed genes were similarly classified in GCs. The first GCs group contains: PTGS2 (prostaglandin-endoperoxide synthase 2), PRKX (protein kinase X-linked), AHI1 (Abelson helper integration site 1), and IL11 (interleukin 11), genes whose expression decreased during culture. The group with the largest increase in expression included ITGA3 (integrin subunit alpha 3), CAV1 (caveolin 1), ANLN (anillin actin-binding protein), and SFRP4 (secreted frizzled-related protein 4). The classification was based on the total analysis of 168 differentially expressed genes in CCs and GCs. During further analysis, among the same 22 common genes belonging to both GCs and CCs, as in the case of separate CCs and GCs analysis, we identified the next two groups of genes: CAV1, ANXA3, ANLN, SFRP4, whose expression was the most up-regulated, and ARF6 and HMGB1, which were down-regulated during culture.

Hierarchical clusterization of the genes is presented as heatmaps in Figure 1.

The enrichment of each GO BP term was calculated as the z-score and is shown on the circular diagrams (Figure 2).

Genes that belonged to one particular GO group also belonged to different categories of GO terms, making it necessary to examine the interactions between selected GO BP terms. To identify the most up and down-regulated genes in CCs and GCs, the mean value of the fold change ratio of each gene between 1, 7, 15 and 30 days of culture was calculated. As a result, the four most up and three most down-regulated genes in CCs were chosen for further analysis. In the case of GCs, the four most up and four most down-regulated genes were evaluated.

The relationship between the selected GO BP terms is presented as a heatmap in Figure 3.

Interactions between the protein products of the 15 differentially expressed genes are presented in Figure 4.

The gene symbols, fold changes in expression, Entrez gene IDs, and corrected *p*-values of 15 genes, which were described and analyzed in CCs and GCs, are shown in Table 2.

TGFBR3, HMGB1, and ARF6 gene expression analyses, examined in CCs during the 30-day cultivation period, revealed a decrease in the expression on the 7th, 15th, and 30th days compared to the 24 h culture. In the case of the HMGB1 gene, its expression decreased over the time course. In the case of SFRP4, ANXA, and DKK1, their expression increase was higher on days 7 and 15. On the 30th day, DKK1 expression was at the highest level. CCND and STC1 genes exhibited a steady increase in expression from the 7th through to the 30th day. HMGB1, CAV1, ANLN, and SFRP4 genes expression, of which was examined both in CCs and GCs, showed a higher expression level in GCs. SFRP4 expression was the highest on day 7. Further analysis of gene expression in GCs showed a uniform increase in ANLN gene expression throughout the culture. It should also be considered that PTGS2 showed the greatest decrease in expression during GC cultivation (Table 2).

Microarray results were validated using quantitative RT-qPCR. Validation was performed separately for two cell types. One gene with the increased expression (TGFBR3) and four genes with the lowest change in expression (STC1, CCND1, DKK1, ANXA3) have been validated in CCs cells. In addition, GC-specific genes were validated: genes with the highest change in expression; PTGS2, PRKX, AHI1, IL11, and lowest change in expression; CAV1, SFRP4. The expression direction of CC-specific genes in all cases was consistent with the direction of expression change in the expression of microarrays except for the DKK1 gene on day 30 of the in vitro primary culture (Figure 5). Directions of change in the expression of all GC-specific genes have been confirmed (Figure 6).

The shape of the GCs and CCs was documented at particular time intervals. CC cells very quickly adhered to the bottom of the culture plate, and their shape was spherical a few hours after seeding (Figure 7). The shape of GC cells changed from star-like cells to more fusiform, fibroblast-like (Figure 8). CC and GC cell shapes were compared at individual time intervals. Both cell types showed a similar change in morphology. The changed GC morphology is referred to as fibroblast-like cells [26,43,44]. CC cells initially showed a spherical shape with long outgrowths. A similar morphology of CC cells in the 2D culture system was obtained by Combelles et al. [45].

## 4. Discussion

Folliculogenesis is a complex and dynamic process of creating follicles that cover the outer layer of the ovary. It belongs to the most important reproductive processes, whose goal is to create an appropriate environment for oocyte growth and maturation [46,47]. According to McCoard et al., primitive, primary, and secondary follicles are all present in the cortical part of the fetal ovary [48]. Primitive follicles are characterized by a single layer of flat cells with the oocyte occupying the center. By transitioning into the primary follicle, the volume of oocyte and adjacent granulosa cells begins to increase. Granulosa cells progressing through the secondary and tertiary follicular phase intensively proliferate, changing shape from flat to cubic, leading to ovulatory Graafian follicle formation with the two distinct GC and CC populations. Due to GC and CC availability as the primary granulosa cells’ form, they have become a representative biological material for understanding their epithelial origin. Furthermore, the GC and CC long-term in vitro culture enabled us to understand their stem-like properties and possible directions of differentiation.

GCs and CCs studied over long-term in vitro cultures may reveal the primary origin and function markers. The current study may be the first step in creating a model of ovarian follicle formation in in vivo conditions.

In cumulus cells, the STC1 (stanniocalcin 1) gene exhibited the highest increase in expression during the 30 days of culture. The expression of this gene is essential for phosphate and calcium ion transport regulation. In addition, this glycoprotein hormone performs paracrine functions and can be found in various types of human tissues [49]. Increased expression of this gene is associated with a number of cancers [50], including ovarian cancer [51]. In immortalized ovarian epithelial cell lines or normal ovarian tissue, Liu et al. demonstrated a higher content of STC1 protein in human ovarian cancer cell lines and ovarian cancer tissue. Cells with increased STC1 gene expression were characterized by faster migration and proliferation, as well as a tendency to form colonies during cultivation [51]. A significant role of this gene has also been found in breast cancer control, development, and progression. Therefore, as a breast cancer genetic marker, it can be used for therapeutic purposes [52]. Stanniocalcin 1 proliferation-promoting mechanism is based on the increased expression of three cyclins: A, B, and E, and Cyclin-dependent kinase 2 (CDK2). CDK2 is also known as cell division protein kinase 2 [53]. Porcine granulosa cells research suggests that by producing O_2_^−^, STC1 adversely affects their redox status. Consequently, by modifying the intracellular production of ROS (reactive oxygen species), STC1 can change GC activity and function [54]. In addition to the paracrine role of the STC1 protein in GCs, its high expression has been reported both in in vivo and in vitro matured oocytes [55], confirming this gene’s importance in the bidirectional communication between the oocyte and cumulus or granulosa cells. CCs exhibit high STC1 gene expression, confirming its significant importance for proper communication with the oocyte. In addition, its overexpression in the long-term culture allows speculation to cause tumor formation. The next gene, CCND1 (cyclin D1), together with CDK4 or CDK6, controls the G1/S cell transition. CCND1 overexpression leads to cell cycle changes and is noted in various types of cancers. Decrease in its expression causes inhibition of cellular proliferation, while increased levels of cyclin 1 are noted in cells that are rapidly dividing. Blocking of the CCND1 function is used to inhibit human granulosa tumor cell proliferation [56]. Presumably, CCND1 and β-catenin (Bcat) can be used together as granulosa cell biomarkers in prehierarchical follicles (PFs) [57]. Activation of CCND1 and ARHGEF7 (rho guanine nucleotide exchange factor 7) through interaction with FOXO3A (forkhead box O3) stimulates cell cycle progression through the G1/S phase in GCs and controls their follicular proliferation [58]. The current results concerning CCND1 show that similar to STC1, these genes exhibit pro-oncogenic properties in proliferating and differentiating CCs. The last two analyzed genes among CCs from the group of those with the highest increase in expression are DKK1 (dickkopf WNT signaling pathway inhibitor 1) and ANXA3 (annexin A3). By binding to the LRP6 (LDL receptor-related protein 6) co-receptor, the protein encoded by DKK1 inhibits the Wnt signaling pathway [59]. Overexpression of this gene is commonly found in cancer cell lines and is responsible for their intense growth, proliferation, and invasiveness. DKK1 expressions have already been found in female and male gonads. In the early stages of fetal ovarian development, the GATA4-FOG2 (GATA binding protein 4-FOG family member 2) transcription complex inhibits its expression [60]. Annexin A3-like DKK1, plays an important role in cancer cell formation and proliferation, as well as in their apoptosis and signaling transmission [61]. DKK2 may play a key role in metastasis promotion and breast cancer development [62], but its role in ovarian tissue has not been defined.

DKK2 expression levels result from the different specificity of expression microarrays in relation to the subsequent validation using RT-qPCR. Different directions of change in the level of expression up-regulation/down-regulation observed for the DKK1 gene in both methods may be the result of the presence of numerous transcript variants. In the case of RT-qPCR, the designed primer pairs may not amplify such a large number of variants, which may be the basis for the observed discrepancies in the results.

Among CC genes exhibiting lower levels, the TGFBR3 (transforming growth factor-beta receptor 3) gene encoding a serine/threonine-protein kinase was the most under-expressed. The current results are consistent with previous work showing reduced levels of TGFBR3 receptor expression in various tumors [63]. TGFBR3 is an inhibin co-receptor that promotes its interaction with ACVR2 (activin A receptor type 2), ACR2B (activin A receptor type 2B), and BMPR2 (bone morphogenetic protein receptor type 2) [64]. Matiller et al. evaluated TGFBR3 expression and found its increased expression in bovine granulosa ovarian cysts previously induced by adrenocorticotropin (ACTH) supplementation [65].

Among the genes analyzed in GCs, the highest expression increase was observed in the SFRP4 (secreted frizzled-related protein 4) gene, which belongs to the main modulators of the Wnt-signaling pathway. Wnt, together with β-catenin, may be involved in GC apoptosis. Wu et al. reported lower SFRP4 expression in PCOS patients [66]. In mice periovulatory follicles and corpus luteum, this gene’s expression was also drastically reduced [67]. Presumably, in the ovulatory ovarian follicle, SFRP4 activity induction may result in final GCs differentiation [68]. The expression of the last two most up-regulated GCs genes was at a comparable level. The elevated expression of CAV1 (caveolin 1) confirms the results obtained by Ożegowska et al., where this gene was attributed with the biggest change in expression during porcine granulosa short-term in vitro culture [69]. In the ovary, this gene’s main function was associated with ovarian primordial follicle (PF) formation [70].

It appears that during the first 24 hours of primary in vitro culture, both CCs and GCs exhibit decreased expression of epithelial biomarkers while restoring mesodermal origin cell functions, which may explain the increased expression of the migration markers ANLN and ITGA3 [71,72].

In the group of assessed and identified GC genes, the PTGS2 (prostaglandin-endoperoxide synthase 2) gene was characterized with by far the highest expression decrease during culture. PTGS2, also known as COX2 (cyclooxygenase 2), is an enzyme that catalyzes the PGE2 (prostaglandin E2) conversion process. PGE2 and its synthesis, associated with an increase in COX2 activity in CCs, plays a key role in their expansion during oocyte maturation and ovulation during folliculogenesis [73]. Some studies indicate that a detailed analysis of this gene’s expression in cumulus cells may be helpful in the evaluation of oocyte development potential [74]. Presumably, reduced expression levels of this gene may indicate poor CC expansion and low oocyte quality. It has been suggested that COX2 under-expression in the Graafian follicle promotes epithelial cells survival in stress conditions during ovulation wound repair [75]. Given the role of COX2 in the PGE2 secretion induction and AKT pathway activation, crucial for the maintenance of cell proliferation and survival process, decreases in COX2 expression during long-term in vitro GCs culture may indicate the beginning of apoptosis. Furthermore, PRKX (protein kinase X-linked), AHI1 (Abelson helper integration site 1), and IL11 (interleukin 11) genes exhibited elevated expression. However, the roles of these genes in in ovarian tissue have not yet been determined. IL-11 cytokine is produced in many types of tissues and exhibits pleiotropic properties. In the human uterus, the highest levels of this gene and its receptors are found in the endometrium. In addition, during implantation, IL-11 plays a part in stromal cell segregation induced by the progesterone wave. Jang et al. noted for the first time that the LH wave stimulates an increase in IL11 expression in preovulatory follicles, which leads to increased progesterone production [76]. Therefore, in addition to the implantation process, this gene plays a key role in steroidogenesis during ovulation. Furthermore, increased IL11 receptor expression has been observed in malignant and benign ovarian tissue tumors [77].

In all of the 148 examined genes, the juxtaposition of their expression in both CCs and GCs resulted in the selection of 22 genes common for these two cell populations. Among them, CAV1, ANXA3, ANLN, and SFRP4 genes showed the highest increase in expression during the 30 days of culture, while two: ARF6, HMGB1, exhibited decreased expression.

CCs and GCs were characterized by an expression profile of genes involved in cellular epithelialization. Detailed gene analysis has partly confirmed these two cell populations’ etiology and origin from ovarian surface epithelium. Furthermore, a comprehensive analysis of the selected genes enabled the recognition of their epithelial nature. In addition, most of the genes appear to be involved in ovarian tumorigenesis. It does not appear that GCs and ovarian stem cells can form primitive ovarian follicles. As it was stated in an article by Xu J. [11] or Auersperg N. [78], at the molecular level, the current research confirms the possible OSE origin of CCs and GCs.

Gene expression profiling in human ovaries suggests that ovarian surface epithelium may be a source of ovarian cancers [79]. Among many types of ovarian tumors, sex cord tumors are believed to be derived from granulosa cells [80]. Throughout folliculogenesis, oocytes play a decisive role in GC’s and CC’s proliferation [81]. After ovulation, these cells survive and retain the ability to transform into luteal cells or cyst-like structures [82]. GC analysis in oocyte-depleted follicles was evaluated on several animal models. Granulosa cells may contribute to epithelial-derived carcinoma formation, which accounts for approximately 5% of human ovarian cancers. These studies indicate that premature loss of reproductive cells leads to the formation of ovarian tumors with many phenotypes. Molecular mechanisms involved in this process have not yet been defined [82]. The current study focused on the behavior of these cells after controlled ovulation in stimulated ovaries, which indicates an increase in GC and CC expression of genes involved in neoplastic processes.

One of the most dangerous ovarian cancers is oophoroma, which affects one or both of the ovaries and uterus glands. The most deadly ovarian cancer is EOC (epithelial ovarian cancer) [83]. Studied genes with an increase in expression and, which are involved in neoplastic processes, after further research, may be used as biomarkers in the screening of the most malignant neoplasms in women. Further analysis of genes with increased expression may be helpful in the diagnosis of ovarian epithelial tumors. The use of early diagnosis of EOC could turn out to be a key factor in patient survival.

A 30-day cell culture protocol provides the best opportunity to understand in vitro cell properties. Long-term cell cultures allow for an accurate understanding of cell properties in new in vitro conditions. The moment 0 (24 h) corresponds approximately to the physiological properties of cells [26,27], while the following days show the changes taking place in the cell. Cell growth and expression analysis in the following days show the changes that occur in the cells. The 7th day defines a short-term culture, the 15th day reflects changes after the first passage, while day 30 is the end of long-term culture. Additionally, the 30-day in vitro culture of CCs and GCs may reveal the cell fate in vitro. Conducting cell cultures without the influence of external regulators and physiological factors allows the study of the differentiation potential of these cells. It also seems interesting to correlate the growth and differentiation of these cells in in vitro conditions with their in vivo behavior. The current study demonstrated that CCs’ and GCs’ long-term in vitro culture might exhibit markers that suggest their ability to differentiate towards ovarian tumors.

In summary, the analysis of genes presented in this article allowed to confirm GCs’ and CCs’ epithelial origin. Moreover, the long-term in vitro culture provides a new perspective to use CC and GC cell lines as in vitro model systems. Understanding the function and behavior of these cell lines in an in vitro culture model could be helpful to understand the physiological processes occurring in the growing ovarian follicles and maturing oocytes. Understanding these processes can be of great importance in tracing how primary follicles recruit for growth and maturation. Despite available methods used in ART (Assisted reproductive technology) procedures, the biological basis of the multiple follicle-stimulation process is still not fully understood. Creating an ovarian follicle model in laboratory conditions would be an unquestionable tool in understanding it. The future challenge to face is to understand the biological mechanism by which hormones used in ovary stimulation protocols affect ovarian follicle growth, granulosa cell differentiation, and finally, oocyte in vivo maturation. The quality of the maturing oocyte in the ovarian follicle is directly dependent on the CCs [3]. Identification of the genes important for oocyte development in CCs and GCs may be useful and used to determine the potential of in vitro fertilized oocytes [16]. Determining the origin of granulosa cells and understanding the processes of their differentiation into CCs and GCs may give us the possibility of more effective ovary stimulation during IVF procedures. It is also possible that the quality of the obtained oocytes and their potential for fertilization may be increased.

## Figures and Tables

**Figure 1 jcm-11-00073-f001:**
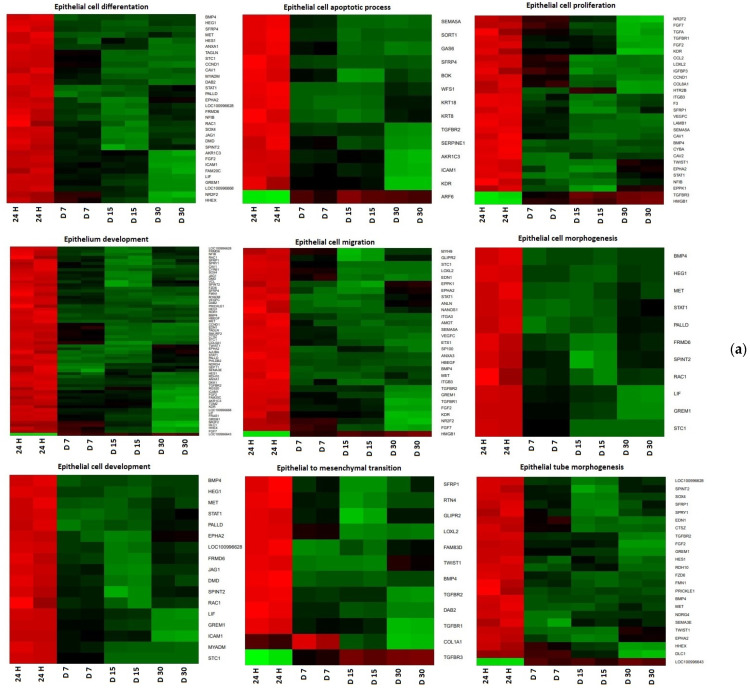
Heat map representation of differentially expressed genes (**a**) CCs; (**b**) GCs. Arbitrary signal intensity derived from microarray analysis is represented by colors (green, higher; red, lower expression). Log 2 signal intensity values for any single gene were resized to Row Z-Score scale (from −2, the lowest expression to +2, the highest expression for a single gene).

**Figure 2 jcm-11-00073-f002:**
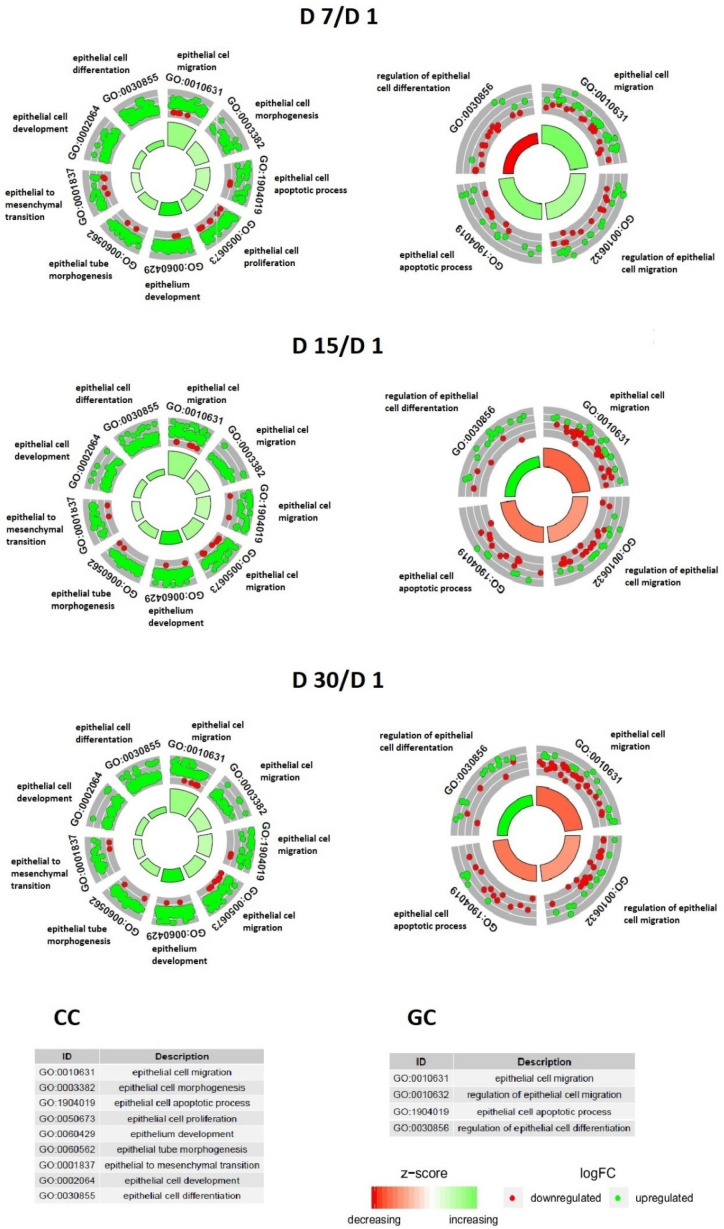
The circle plot showing the differentially expressed genes and z-scores. The outer circle represents a scatter plot for each term of the fold change of the assigned genes. Green circles display up-regulated genes, and red circles display down-regulated genes. The inner-circle illustrates the z-score of each GO BP term. The width of each bar corresponds to the number of genes within the GO BP term, and the color corresponds to the z-score.

**Figure 3 jcm-11-00073-f003:**
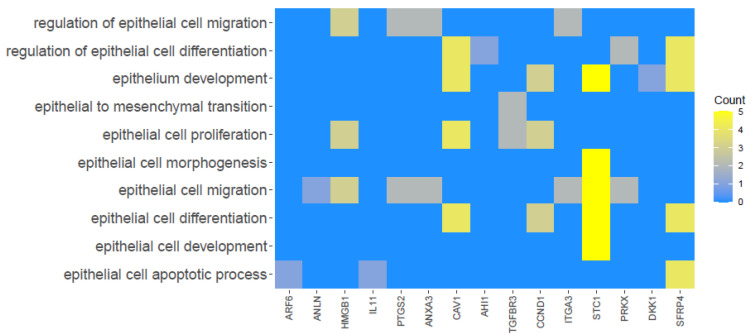
Heatmap illustrating 15 genes based upon the selected GO BP terms. The yellow color is associated with genes relative to the GO Term. The intensity of the color corresponds to the amount of GO BP terms.

**Figure 4 jcm-11-00073-f004:**
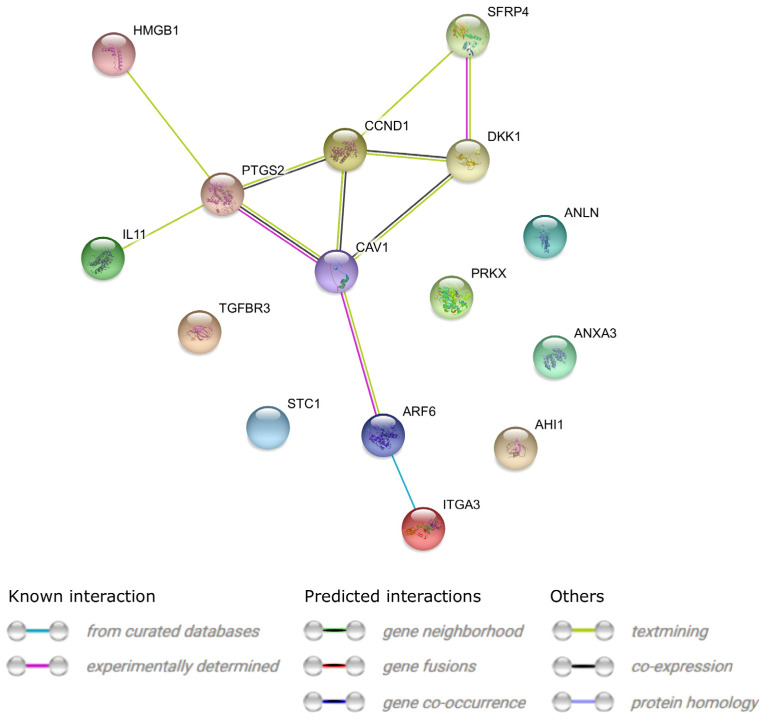
STRING-generated interaction occurrence between 15 chosen differentially expressed genes that belong to the selected GO BP terms. The intensity of the edges reflects the strength of the interaction score.

**Figure 5 jcm-11-00073-f005:**
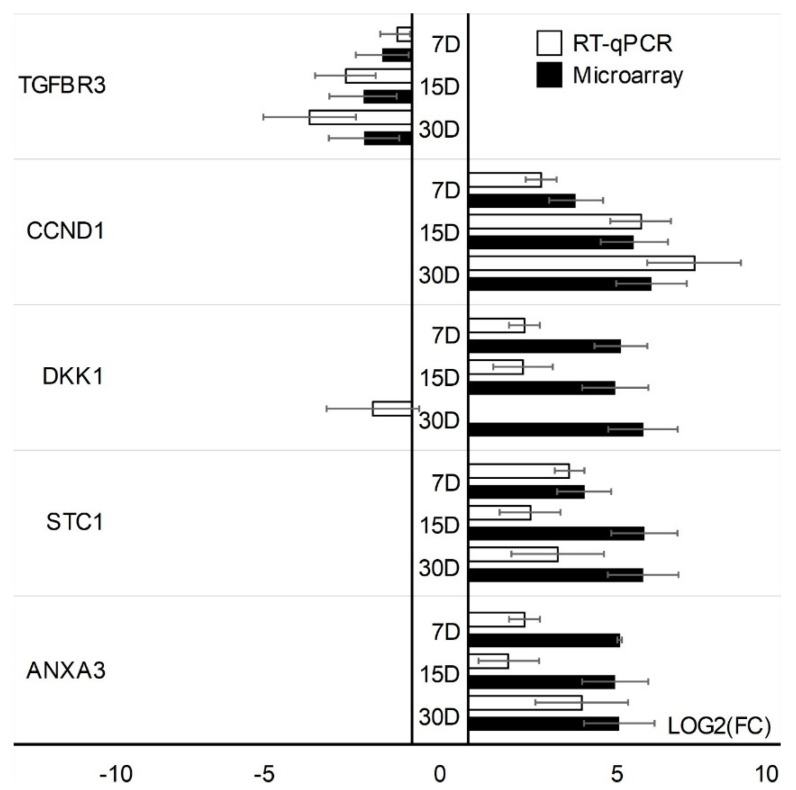
RT−qPCR validation of microarray change expressions (log(FC)) in CCs. Validation of the microarray method was performed in three separate biological repetitions. Each of the biological repetitions was performed in three technical repetitions; D: day of culture; FC: fold change.

**Figure 6 jcm-11-00073-f006:**
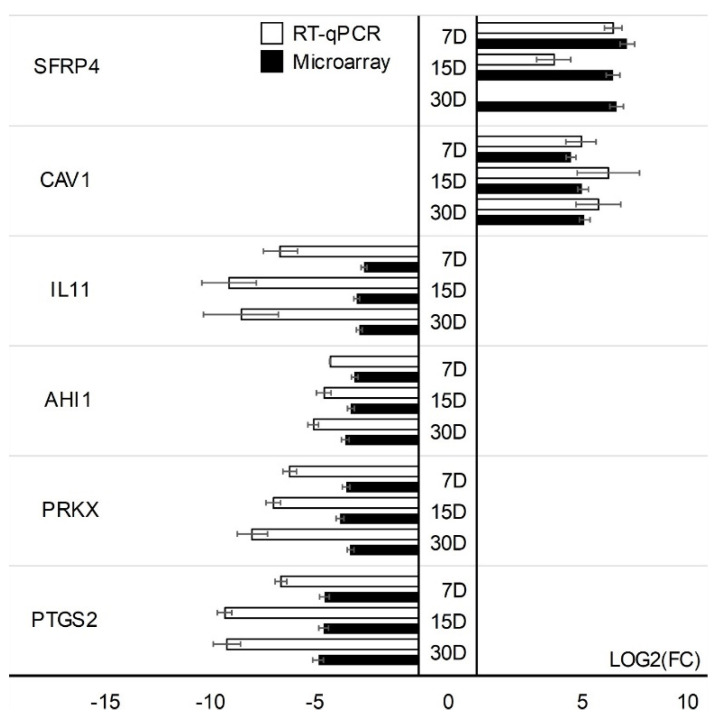
RT−qPCR evaluation of selected genes based upon the microarray data (log(FC)) in GCs. Validation of the microarray method was performed in three separate biological repetitions. Each of the biological repetitions was performed in three technical repetitions; D: day of culture; FC: fold change.

**Figure 7 jcm-11-00073-f007:**
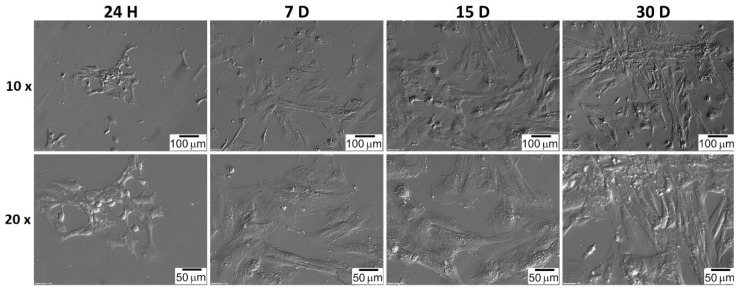
CC’s morphology over 30 days in vitro culture. H: hour of culture, D: day of culture.

**Figure 8 jcm-11-00073-f008:**
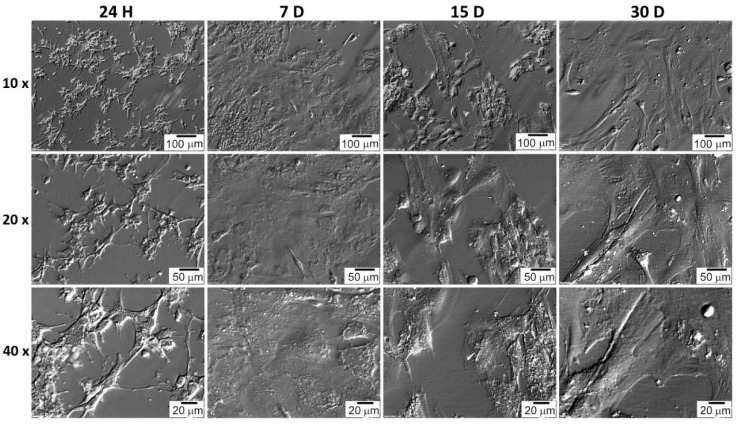
GC’s morphology over 30 days in vitro culture. H: hour of culture, D: day of culture.

**Table 1 jcm-11-00073-t001:** Oligonucleotide sequences of primers used for RT−qPCR analysis.

Gene	Type of Cells	Primer Sequence (5′-3′)	Product Size (bp)	Entrez Gene ID (on Accession)
TGFBR3	CCs	CCAAGATGAATGGCACACAC CCATCTGGCCAACCACTACT	151	7049
ANXA3	CCs	GTTGGACACCGAGGAACAGT CACTAGGGCCACCATGAGAT	249	306
DKK1	CCs	TCCGAGGAGAAATTGAGGAA CCTGAGGCACAGTCTGATGA	157	22,943
CCND1	CCs	GAGGAAGAGGAGGAGGAGGA GAGATGGAAGGGGGAAAGAG	236	595
STC1	CCs	TGATCAGTGCTTCTGCAACC GACGAATGCTTTTCCCTGAG	242	6781
PTGS2	GCs	TGAGCATCTACGGTTTGCTG TGCTTGTCTGGAACAACTGC	158	5743
PRKX	GCs	CACGGGGCTCTTCTACTCTG CTACCAGCTTCTTGGCGAAC	155	5613
AHI1	GCs	TTGGAACCCAGAAACAGGAG GGGCATCTTGACTTTGGTGT	239	54,806
IL11	GCs	GCTGCACCTGACACTTGACT CACCCCTGCTCCTGAAATAA	249	3859
CAV1	GCs	TCTCTACACCGTTCCCATCC CAATCTTGACCACGTCATCG	164	857
SFRP4	GCs	GCCTGGGACAGCCTATGTAA TCTGTACCAAAGGGCAAACC	160	6424
GAPDH	CCs, GCs	TCAGCCGCATCTTCTTTTGC ACGACCAAATCCGTTGACTC	90	2597
ACTB	CCs, GCS	AAAGACCTGTACGCCAACAC CTCAGGAGGAGCAATGATCTTG	132	60
HPRT	CCs, GCs	TGGCGTCGTGATTAGTGATG ACATCTCGAGCAAGACGTTC	141	3251

**Table 2 jcm-11-00073-t002:** Fold change in expression ratio, Entrez gene IDs, corrected *p*-values, and mean values of the fold change ratio of the 15 selected genes expressed in CCs and GCs.

Gene Symbol	Entrez Gene ID	Ratio 7 d/24 h	Ratio 15 d/24 h	Ratio 30 d/24 h	Adj. *p*-val. 7 d/24 h	Adj. *p*-val. 15 d/24 h	Adj. *p*-val. 30 d/24 h	Mean Ratio
CCs								
TGFBR3	7049	−1.998	−3.083	−3.056	4.62 × 10^−5^	1.29 × 10^−6^	1.04 × 10^−6^	−2.7125
HMGB1	3146	−2.054	−2.261	−2.526	6.96 × 10^−6^	2.04 × 10^−6^	6.84 × 10^−7^	−2.2806
ARF6	382	−2.044	−2.389	−2.163	1.93 × 10^−5^	3.58 × 10^−6^	6.57 × 10^−6^	−2.1993
LOC100 996643	25902	−2.042	−2.005	−2.167	1.04 × 10^−5^	8.36 × 10^−6^	3.38 × 10^−6^	−2.0717
CAV1	857	3.402	5.865	5.26	2.02 × 10^−6^	1.30 × 10^−7^	1.39 × 10^−7^	4.8427
ANLN	54443	5.724	5.910	3.1	4.27 × 10^−7^	2.57 × 10^−7^	4.09 × 10^−6^	4.912
SFRP4	6424	22.184	28.775	27.230	5.34 × 10^−8^	1.93 × 10^−8^	1.44 × 10^−8^	26.0633
ANXA3	306	34.272	30.545	33.593	5.51 × 10^−9^	3.16 × 10^−9^	1.57 × 10^−9^	32.8035
DKK1	22943	34.811	30.492	58.943	5.58 × 10^−9^	3.43 × 10^−9^	9.21 × 10^−10^	41.4158
CCND1	595	12.080	46.968	70.869	5.54 × 10^−9^	6.96 × 10^−10^	4.92 × 10^−10^	43.3059
STC1	6781	14.962	60.004	58.346	5.58 × 10^−9^	6.96 × 10^−10^	4.92 × 10^−10^	44.4377
GCs								
PTGS2	5743	−22.653	−23.248	−27.601	9.56 × 10^−4^	8.93 × 10^−4^	6.17 × 10^−4^	−24.5012
PRKX	5613	−11.071	−13.497	−9.694	2.46 × 10^−3^	1.83 × 10^−3^	2.20 × 10^−3^	−11.4212
AHI1	54806	−8.453	−9.569	−11.378	2.36 × 10^−3^	1.83 × 10^−3^	1.19 × 10^−3^	−9.80055
IL11	3589	−6.124	−7.771	−7.132	2.11 × 10^−3^	1.35 × 10^−3^	1.19 × 10^−3^	−7.00951
ARF6	382	−2.201	−2.595	−1.954	2.70 × 10^−2^	1.34 × 10^−2^	3.60 × 10^−2^	−2.25027
HMGB1	3146	1.889	2.106	1.967	4.38 × 10^−2^	2.45 × 10^−2^	2.96 × 10^−2^	1.98783
ANXA3	306	13.404	19.427	21.684	2.57 × 10^−2^	1.52 × 10^−2^	1.21 × 10^−2^	18.1721
ITGA3	3675	16.927	18.842	22.47	2.77 × 10^−2^	2.23 × 10^−2^	1.65 × 10^−2^	19.4133
CAV1	857	22.409	32.071	34.618	1.95 × 10^−3^	1.33 × 10^−3^	9.44 × 10^−4^	29.6998
ANLN	54443	59.051	63.936	59.669	1.57 × 10^−3^	1.34 × 10^−3^	1.09 × 10^−3^	60.8857
SFRP4	6424	141.911	89.682	101.05	5.14 × 10^−3^	6.31 × 10^−3^	5.01 × 10^−3^	110.8814

## Data Availability

All analyzed microarray data are available and it is possible to download them from the GEO database. CCs: https://www.ncbi.nlm.nih.gov/geo/query/acc.cgi?acc=GSE149033 (accessed on 19 September 2019) and GCs: https://www.ncbi.nlm.nih.gov/geo/query/acc.cgi?acc=GSE129919 (accessed on 19 September 2019).

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
