# Peer review of "Cellular Processes in Human Ovarian Follicles Are Regulated by Expression Profile of New Gene Markers—Clinical Approach"

_jcm, 2021, doi:10.3390/jcm11010073_

Round 1
Reviewer 1 Report
The publication by Chermuła et al. presents expression profile of new gene markers in human cumulus (CCs) and granulosa (GCs) cells. The research idea and obtained results are very interesting and seem to be very important for understanding the mechanism of cellular processes in human ovarian follicles.
I hope that my comments will help to better understand the text.
- The summary is well written and outlining the main contributions and aim of the paper.
- In the introduction I found inaccuracies that disturb the understanding of the text. Some sentences are not clear enough:
- ”Proper understanding of the biological origin of granulosa cells is the basis for understanding the structure and function of granulosa cells, ovarian follicles and the entire ovary.” – it should be described in more detail
- “In these studies and those analyzing the expression of genes in porcine oocytes before and after their in vitro maturation [22] microarray expression techniques were used” - Do the authors mean the research in citation 22? It need to be clarify.
- At lines 95-100 as well as 104-107 the references are missing. It should be added, especially when it comes to incubation times.
- The methodology and obtained results are described in a concise and understandable way.I have no objections to this part of the paper.
- The authors discuss all the results in a logical way based on relevant citations from previous studies. However, as in the introduction, some parts of the text are unnecessary and should be rewritten, because it impairs understanding, e. g.: sentence at lines 477-479 would sound better at this form: “Matiller et al. evaluated TGFBR3 expression and found its increased expression in bovine’ granulosa ovarian cysts previously induced by adrenocorticotropin (ACTH) supplementation [58]”. Please review and edit similar inaccuracies.
- Other comments:
- The references require the addition of several recent publications which have been published in the last 5 years.
- Abbreviations in the text should be standardized.For example, in the introduction, the abbreviation or the full name appears in the brackets: “mural granulosa cells (GCs) and cumulus cells (CCs)”, but “FSH (follicle stimulating hormone) and IGF1 (insulin like growth factor 1)” or “the OSE (ovarian surface epithelium)” - please make it uniform. Similar errors appear throughout the text.
Author Response
Poznan, December 18th, 2021
Dear Reviewer,
Thank you very much for the careful review of our manuscript entitled: „Cellular processes in human ovarian follicles are regulated by expression profile of new gene markers - clinical approach” We have significantly revised the manuscript, and our response to Reviewer #1 is described below.
Response to Reviewer’s Comments
REVIEWER 1
(comments replies and manuscript changes are highlighted in red)
Comments and Suggestions for Authors
The publication by Chermuła et al. presents expression profile of new gene markers in human cumulus (CCs) and granulosa (GCs) cells. The research idea and obtained results are very interesting and seem to be very important for understanding the mechanism of cellular processes in human ovarian follicles.
I hope that my comments will help to better understand the text.
Thank you very much for the positive feedback and appreciation of our research innovations.
Answers to the reviewer's questions and suggestions:
The summary is well written and outlining the main contributions and aim of the paper.
In the introduction I found inaccuracies that disturb the understanding of the text. Some sentences are not clear enough:
”Proper understanding of the biological origin of granulosa cells is the basis for understanding the structure and function of granulosa cells, ovarian follicles and the entire ovary.” – it should be described in more detail
This sentence has been modified:
„Confirmation of granulosa cells origin may help to approximate these cells function in the ovarian follicle structure and the entire ovary. These cells are essential for oocyte maturation during folliculogenesis, Graffian follicle ovulation and corpus luteum formation [16]. The negative aspect of these cells may be the possibility of their transformation into cancer cells [17].”
“In these studies and those analyzing the expression of genes in porcine oocytes before and after their in vitro maturation [22] microarray expression techniques were used” - Do the authors mean the research in citation 22? It need to be clarify.
This sentence has been modified too:
„Microarry expression technology was used following previous studies published by Ożegowska et al. [22], which analyze gene expression in porcine oocytes before and after maturation."
At lines 95-100 as well as 104-107 the references are missing. It should be added, especially when it comes to incubation times.
Missing references were inserted.
[16] Chermuła B, Kranc W, Jopek K, Budna-Tukan J, Hutchings G, Dompe C, et al. Human Cumulus Cells in Long-Term In Vitro Culture Reflect Differential Expression Profile of Genes Responsible for Planned Cell Death and Aging-A Study of New Molecular Markers. Cells 2020. https://doi.org/10.3390/cells9051265.
[25] Kranc W, BrÄ…zert M, Celichowski P, Bryja A, Nawrocki MJ, Ożegowska K, et al. ‘Heart development and morphogenesis’ is a novel pathway for human ovarian granulosa cell differentiation during long-term in vitro cultivation-a microarray approach. Mol Med Rep 2019;19:1705–15. https://doi.org/10.3892/mmr.2019.9837.
[26] Kossowska-Tomaszczuk K, De Geyter C, De Geyter M, Martin I, Holzgreve W, Scherberich A, et al. The Multipotency of Luteinizing Granulosa Cells Collected from Mature Ovarian Follicles. Stem Cells 2009. https://doi.org/10.1634/stemcells.2008-0233.
[28] Kranc W, Budna J, Kahan R, ChachuÅ‚a A, Bryja A, CiesióÅ‚ka S, et al. Molecular basis of growth, proliferation, and differentiation of mammalian follicular granulosa cells. J Biol Regul Homeost Agents 2017.
The methodology and obtained results are described in a concise and understandable way.I have no objections to this part of the paper.
The authors discuss all the results in a logical way based on relevant citations from previous studies. However, as in the introduction, some parts of the text are unnecessary and should be rewritten, because it impairs understanding, e. g.: sentence at lines 477-479 would sound better at this form: “Matiller et al. evaluated TGFBR3 expression and found its increased expression in bovine’ granulosa ovarian cysts previously induced by adrenocorticotropin (ACTH) supplementation [58]”. Please review and edit similar inaccuracies.
The manuscript has been revised following the reviewer’s suggestions.
Other comments:
The references require the addition of several recent publications which have been published in the last 5 years.
As suggested, the article now cites publications which have been published in the last 5 years.
[2] Li L, Shi X, Shi Y, Wang Z. The Signaling Pathways Involved in Ovarian Follicle Development. Front Physiol 2021. https://doi.org/10.3389/fphys.2021.730196.
[3] Turathum B, Gao EM, Chian RC. The function of cumulus cells in oocyte growth and maturation and in subsequent ovulation and fertilization. Cells 2021. https://doi.org/10.3390/cells10092292.
[17] Schmid N, Dietrich KG, Forne I, Burges A, Szymanska M, Mei-Dan R, et al. Sirtuin 1 and sirtuin 3 in granulosa cell tumors. Int J Mol Sci 2021. https://doi.org/10.3390/ijms22042047.
[28] Kranc W, Budna J, Kahan R, ChachuÅ‚a A, Bryja A, CiesióÅ‚ka S, et al. Molecular basis of growth, proliferation, and differentiation of mammalian follicular granulosa cells. J Biol Regul Homeost Agents 2017.
[54] Jepsen MR, Kløverpris S, Bøtkjær JA, Wissing ML, Andersen CY, Oxvig C. The proteolytic activity of pregnancy-associated plasma protein-A is potentially regulated by stanniocalcin-1 and -2 during human ovarian follicle development. Hum Reprod 2016. https://doi.org/10.1093/humrep/dew013.
[60] Andre P, Song H, Kim W, Kispert A, Yang Y. Wnt5a and Wnt11 regulate mammalian anterior-posterior axis elongation. Dev 2015. https://doi.org/10.1242/dev.119065.
[84] Perales-Puchalt A, Svoronos N, Rutkowski MR, Allegrezza MJ, Tesone AJ, Payne KK, et al. Follicle-stimulating hormone receptor is expressed by most ovarian cancer subtypes and is a safe and effective immunotherapeutic target. Clin Cancer Res 2017. https://doi.org/10.1158/1078-0432.CCR-16-0492.
Abbreviations in the text should be standardized. For example, in the introduction, the abbreviation or the full name appears in the brackets: “mural granulosa cells (GCs) and cumulus cells (CCs)”, but “FSH (follicle stimulating hormone) and IGF1 (insulin like growth factor 1)” or “the OSE (ovarian surface epithelium)” - please make it uniform. Similar errors appear throughout the text.
The abbreviations in the text have been unified.
With kind regards,
Yours sincerely,
Bartosz Kempisty PhD,
Department of Anatomy
Department of Histology and Embryology,
University of Medical Sciences
Poznan, Poland
Reviewer 2 Report
This manuscript presents an in depth genetic analysis of origin an behaviour
of ovarian follicular granulosa and cumulus cells.
This precisely done lab work using available technology could identify a new genetic marker panel comprising mechanisms of up and down regulation of ovarian epithelian cells.
From a clinical perspective it would be of importance to outline the diagnostic potential of these findings. In two different directions:
- The significance and value for the screening of ovarian cancer.
- The significance for improvement of ovarian stimulation to better yield oocytes with a higher capacity towards fertilization and pregnancy likelyhood.
Author Response
Poznan, December 18th, 2021
Dear Reviewer,
Thank you very much for the careful review of our manuscript entitled„Cellular processes in human ovarian follicles are regulated by expression profile of new gene markers - clinical approach”. We have significantly revised the manuscript, and our response to the second reviewer is below.
Response to Reviewer’s Comments
REVIEWER 2
(comments replies and manuscript changes are highlighted in blue)
The spelling check has been carried out.
Comments and Suggestions for Authors
Here are the answers to the Reviewer's comments:
This manuscript presents an in depth genetic analysis of origin an behaviour
of ovarian follicular granulosa and cumulus cells.
This precisely done lab work using available technology could identify a new genetic marker panel comprising mechanisms of up and down regulation of ovarian epithelian cells.
Thank you very much for the positive feedback and appreciation of our research.
From a clinical perspective it would be of importance to outline the diagnostic potential of these findings. In two different directions:
As suggested, the diagnostic potential of the presented studies was presented in the discussion section:
- The significance and value for the screening of ovarian cancer.
„One of the most dangerous ovarian cancers is oophoroma, affects one or both of the ovaries and uterus glands. The most deadly ovarian cancer is EOC (epithelial ovarian cancer) [84]. Studied genes with an increase in expression and, which are involved in neoplastic processes, after further research, may be used as biomarkers in screening of the most malignant neoplasms in women. Further analysis of genes with increased expression may be helpful in the diagnosis of ovarian epithelial tumors. The use of early diagnosis of EOC could turn out to be a key factor in patient survival.”
- The significance for improvement of ovarian stimulation to better yield oocytes with a higher capacity towards fertilization and pregnancy likelyhood.
„The quality of the maturing oocyte in the ovarian follicle is directly dependent on the CCs [3]. Identification of genes important for oocyte development in CCs and GCs may be useful and used to determine the potential of in vitro fertilized oocytes [16]. Determining the origin of granulosa cells and understanding the processes of their differentiation into CCs and GCs may give us the possibility of more effective ovaries stimulation during IVF procedures. It is also possible that the quality of the obtained oocytes and their potential for fertilization may be increased.”
Both of these approaches indicate the direction in which subsequent research on these genes should go in the future.
With kind regards,
Yours sincerely,
Bartosz Kempisty PhD,
Department of Anatomy
Department of Histology and Embryology,
University of Medical Sciences
Poznan, Poland

Reviewer 3 Report
Manuscript summary: This manuscript utilised long-term in vitro culture and microarray expression analysis to investigate differential gene expression between cumulus and granulosa cells obtained from human ovarian follicles. Several key genes were differentially expressed, providing insight about the epithelial origin of these cell types and their potential contributions to ovarian tumorigenesis. The manuscript is fairly well written with appropriate experimental design and methodology employed. After the first revision, the overall quality of the manuscript has improved, and some improvements to discussion of the biological significance, future directions and clinical relevance of this research have been made. Comments and suggestions are below.
Several comments from my first revision were either not addressed or only partially addressed:
- Please further increase the font size and resolution clarity of Figures 1 & 2, these are still too small to read clearly and are pixelated in the PDF document I was provided with.
- Figure 5 & 6 – Were individual or pooled samples validated against the microarray using RT-qPCR? If pooled samples were used, why? Please clarify this in the text or figure legend.
- Why was DKK1 expression significantly different between the microarray and RT-qPCR? Please discuss this in the discussion.
- How does the growth and differentiation of these cells in vitro correlate to their in vivo behaviour? Please expand on this in the discussion.
However, considering the feedback from the other reviewer during the first submission, I have some concerns about the significance, novelty and soundness of this study. I agree with them that it seems this group tend to apply this same methodology (i.e. collect human granulosa/cumulus cells from IVF patients, culture them long-term & perform microarray/qPCR analysis of different genes depending on the research question) in many of their publications (Examples: 1, 2, 3, 4, 5). It appears that they always seem to find differences in gene expression after long-term culture which conveniently align with the research question of that manuscript, for example that human granulosa cells display features of osteoblasts, muscle lineage, neuronal characteristics, etc. It is interesting that they can always consistently find completely different results to support their research questions using essentially the same cells in the same protocol. So although I don't see any major flaws with the current study, the fact that this same protocol has essentially been "rinsed and repeated" many times does unfortunately raise some concerns about the scientific soundness and robustness of this manuscript.
Author Response
Poznan, December 18th, 2021
Dear Reviewer,
Thank you very much for the careful review of our manuscript. We have significantly revised the manuscript entitled: „Cellular processes in human ovarian follicles are regulated by expression profile of new gene markers - clinical approach” Below are our detailed responses to the reviewer’s queries.
Response to Reviewer’s Comments
REVIEWER 3
(comments replies and manuscript changes are highlighted in green)
The spelling check has been carried out.
Comments and Suggestions for Authors
Manuscript summary: This manuscript utilised long-term in vitro culture and microarray expression analysis to investigate differential gene expression between cumulus and granulosa cells obtained from human ovarian follicles. Several key genes were differentially expressed, providing insight about the epithelial origin of these cell types and their potential contributions to ovarian tumorigenesis. The manuscript is fairly well written with appropriate experimental design and methodology employed. After the first revision, the overall quality of the manuscript has improved, and some improvements to discussion of the biological significance, future directions and clinical relevance of this research have been made. Comments and suggestions are below.
Thank you for appreciating and accepting the changes which were made after the first revision. I attach the answers to the individual sections of the Reviewer's comments. This manuscript version has been corrected and changed as Reviewer suggested.
Several comments from my first revision were either not addressed or only partially addressed:
Please further increase the font size and resolution clarity of Figures 1 & 2, these are still too small to read clearly and are pixelated in the PDF document I was provided with.
In line with the reviewer's recommendations, figures 1 and 2 have been modified and presented in a new, more readable way. New figures in the current form are legible in a Word or PDF file. Unfortunately, the effect of the review system is not known until after submission.
Figure 5 & 6 – Were individual or pooled samples validated against the microarray using RT-qPCR? If pooled samples were used, why? Please clarify this in the text or figure legend.
Material & methods section has been revised:
„The individual samples have been used in microarray validation. To eliminate technical errors related to the application of reagents to a 96-well plate, each biological repetition was performed in 3 technical repetition.”
Figure 5 and 6 legend have been supplemented:
„Validation of the microarray method was performed in 3 separate biological repetitions. Each of the biological repetitions was performed in 3 technical repetitions.”
Why was DKK1 expression significantly different between the microarray and RT-qPCR? Please discuss this in the discussion.
Missing explanation was added to the discussion section:
„DKK2 expression level results from the different specificity of expression microarrays in relation to the subsequent validation using RT-qPCR. Different direction of change in the level of expression upregulation/downregulation observed for the DKK1 gene in both methods used to evaluate expression may be the result of the presence of numerous transcript variants. In the case of RT-qPCR, the designed primer pairs may not amplify such a large number of variants, which may be the basis for the observed discrepancies in the results.”
How does the growth and differentiation of these cells in vitro correlate to their in vivo behaviour? Please expand on this in the discussion.
Discussion has been supplemented by:
“A 30-day cell culture protocol, provides the best opportunity to understand in vitro cell properties. Long term cell culture allows for accurate understanding of cell properties in new in vitro conditions. The moment 0 (24h) corresponds approximately to the physiological properties of cells [27,29], while the following days show the changes taking place in the cell. Cell growth and expression analysis in the following days show the changes that occur in the cells. The 7th day defines short-term culture, the 15th day reflects changes after the first passage, while day 30 is the end of long-term culture. Additionally, the 30-day in vitro culture of CCs and GCs may reveal cell fate in vitro. Conducting cell culture without the influence of external regulators and physiological factors allows study regarding the differentiation potential of these cells. It also seems interesting to correlate growth and differentiation of these cells in in vitro conditions with their in vivo behavior. The current study demonstrated that CCs and GCs long-term in vitro culture, may exhibit markers that suggest their ability to differentiate towards ovarian tumors.”
However, considering the feedback from the other reviewer during the first submission, I have some concerns about the significance, novelty and soundness of this study. I agree with them that it seems this group tend to apply this same methodology (i.e. collect human granulosa/cumulus cells from IVF patients, culture them long-term & perform microarray/qPCR analysis of different genes depending on the research question) in many of their publications (Examples: 1, 2, 3, 4, 5). It appears that they always seem to find differences in gene expression after long-term culture which conveniently align with the research question of that manuscript, for example that human granulosa cells display features of osteoblasts, muscle lineage, neuronal characteristics, etc. It is interesting that they can always consistently find completely different results to support their research questions using essentially the same cells in the same protocol. So although I don't see any major flaws with the current study, the fact that this same protocol has essentially been "rinsed and repeated" many times does unfortunately raise some concerns about the scientific soundness and robustness of this manuscript.
Our team's research has been associated with reproduction and development of humans and animals for many years. We focus primarily on the properties of GCs and CCs cells in vitro. Recent studies on GCs and CCs maintained in a long-term primary in vitro culture indicate a change in the properties of GCs in vitro, their prolonged viability, as well as the potential for differentiation into other cell types.
Moreover, responding to the reviewer's remark, it is important to emphasize that the aim of the experiments carried out within this publication was clearly defined and referred to the processes that human GCs and CCs cells undergo during long-term primary culture. These processes have not yet been extensively described in the literature and there is a lack of data on new molecular markers, in the form of protein coding genes, responsible for the regulation of these processes at the cellular level.
Our current research is based on completely new analyses, which compare GCs and CCs transcriptomic profile. Results presented in this article, refer of both cell types (GCs and CCs) morphogenetic origin. Their epithelial nature has not been previously considered in detail. Global transcriptomic screening analysis and profiling, allowed us to identify new genetic markers. Their protein products may be involved in epithelial mesenchymal transition process, during GCs and CCs differentiation into other cell types, indicating their parental nature / properties. Such behaviour pointing out to their stem cell character and properties.
Moreover, GCs and CCs individual genes expression profile is different, as these cells during long-term culture may differentiate into other cell types. Both cell types similarily lose their original properties (if hormone supplementation was not used in the culture. Specifically the cells , exhibit decreased expression of FSH and LH receptors). Cells which were used in our study were not stimulated with any supplements. Cells collected from patients belonging to a similar clinical group and kept in the same in vitro primary culture conditions may show the same genomic properties. However, belonging to given ontological groups genes individual transcriptomic profile and expression differs significantly, which suggests their different influence on regulated processes defined by given "ontological groups". The experimental approach allowed the identification of several ontological groups including genes regulating processes atypical for GCs and CCs cells.
The results obtained from expression microarrays (Affymetrix) typically generate hundreds of functional groups (ontology groups) that, using appropriate biostatistical tools (e.g., cut-off at the "fold=2" level), allow the identification of thousands of different transcripts (in the case of the cells analysed, this is over 12,000 different mRNAs). Functional groups include genes whose protein products are responsible for various mechanisms and processes in cells. The most important step in carrying out the microarray analysis is to group the generated functional groups including genes with similar functions. Subsequently it is possible to analyse the obtained results in terms of regulation of biological processes occurring in the studied cells. Often the functional groups are not related to each other in any way, therefore it is not possible to combine the obtained results in common analyses.
Therefore, the publication of our team cited by the second reviewer from the first revision (https://www.mdpi.com/2077-0383/9/6/2006) is in no way related to the results described in the current article, because it concerns strictly defined processes, i.e., expression of new genetic markers responsible for morphogenesis, structure and development, and differentiation of muscle cells. The conclusion of this study is that human GCs cells can differentiate into muscle cells. Thus, both articles address quite different biological processes that human GCs cells undergo when subjected to long-term in vitro culture, and any elements of convergence are due to the use of similar biostatic tools and studies, covered in the Materials and Methods section. Both articles are all the more valuable as they identify new molecular markers regulating biological processes in human GCs that have not yet been reported in the literature, e.g., expression of muscle cell-specific genes in human GCs and expression of epithelialization markers. Furthermore, both articles address (1) the genetic "origin" of human GCs cells, and (2) their previously undescribed potential to differentiate.
With kind regards,
Yours sincerely,
Bartosz Kempisty PhD,
Department of Anatomy
Department of Histology and Embryology,
University of Medical Sciences
Poznan, Poland

Round 2
Reviewer 3 Report
Thank you for addressing all comments thoroughly.